# The Importance of Place Attachment in the Understanding of Ageing in Place: “The Stones Know Me”

**DOI:** 10.3390/ijerph192417052

**Published:** 2022-12-19

**Authors:** Irene Lebrusán, M. Victoria Gómez

**Affiliations:** 1Centro Internacional Sobre el Envejecimiento (CENIE), Universidad de Salamanca, 37002 Salamanca, Spain; 2Facultad de Ciencias Sociales y Jurídicas, Universidad Carlos III de Madrid, 28903 Madrid, Spain

**Keywords:** ageing in place, place attachment, intimate strangers

## Abstract

As academic literature has shown, there is a preference among older adults to experience old age independently, in their own homes, giving shape to what has been called ageing in place. This phenomenon links residence, life cycle, and the experience of old age. Although it depends on many factors (housing characteristics, the elderly’s economy, or their social support, among others) it is based on place attachment as a key aspect, which comprises two different but interwoven dimensions: the home (private space) and the neighbourhood (social space), understood as an extended sphere of the home. Despite its importance, and beyond the consensus that the time spent in a place increases attachment to it, the processes whereby place attachment is constructed by the elderly and the role of the experience of neighbourhood are little known. This article intended to delve into the meaning of place attachment, its importance, and how it is built by the elderly population in urban areas. For this purpose, a qualitative study was conducted in Spain, selecting Madrid (the biggest city in the country) as a case study of how place attachment is shaped in an urban setting. To obtain in-depth information, data were collected through ethnographic interviews with 37 people aged 65–95 and 1 focus group among elderly aged 65–71. The most relevant finding of the qualitative analysis is the emotional significance of attachment to the local space as an element that enables continuity—as opposed to the rupture to which we associate old age- in the experience of the life cycle of the elderly. It should be noted that some of the results are conditioned by the specificities of Spanish society, particularly with regard to family ties. Potential extrapolations to other realities should take this point into account.

## 1. Introduction—A Place of My Own: Why Ageing in Place Is Important in the Construct and Experience of Ageing

As academic literature has shown, there is a preference among older adults to experience old age independently, in their own homes, giving shape to what has been called ageing in place, both in Spain and in other countries [1,2]. Ageing in place links residence, life cycle, and the experience of old age. The importance of this phenomena is increased by the fact that the elderly who remain at home have a longer life and enjoy greater objective well-being than those who reside in group facilities, even when they are in a situation of dependency [3]. On the other hand, keeping the elderly in their homes has been found to be a cost-cutting policy strategy [4].

Although ageing in place depends on many factors (housing characteristics, the elderly’s economy, or their social support, among others), it is based on place attachment as a key aspect, which in the urban realm comprises two different but interwoven dimensions: the home (private space) and the neighbourhood (social space), which is understood as an extended sphere of the home. 

Building on the great relevance that ageing in place has for the older population, both in terms of the home itself and in terms of the immediate residential environment, this article intends to delve into the meaning of place attachment, its importance, and how it is constructed by the elderly in urban areas, given the fact that most of the population will grow old in cities [5]. With this goal in mind, the qualitative methodology is the proper way to obtain the information. 

The paper is organised as follows: This first section describes the importance of ageing in place as well as its connection to place attachment. Section 2 describes our case study and the materials and methods used in this analysis. Section 3 presents the results of the qualitative analysis. The discussion of the results is shown in Section 4. Finally, Section 5 presents the conclusions and the main lines of future work.

### 1.1. Ageing in Place

Ageing in place is a broad concept open to many nuances, subject to debate and sometimes overused, which makes it difficult to delimit [6]. According to Harris, it is defined as “the effect of time on a population that does not move; remaining in the same residence where a person has spent his or her previous years” [7]. Phillips, Ajrouch and Hillcoat-Nallétamby synthesise it as “getting old without having to move from home” [8]. Certain conceptions assume that the mere fact of living a prolonged time in a place before the end of old age (death) would be conceptually equivalent to ageing in place, and some definitions refer only to the changes that occur in the inhabitants over time and not to the nature of the change on the environment, forgetting that housing is not static [6] and that the relationship between the person and the environment is dynamic [9,10]. This is important in terms of the household changes, but also in terms of the space in which one lives, and the way in which it is experienced, as well as the relationship between home and space.

For Lawton [11], ageing in place would be a transaction between the person and his or her environment, characterised by changes in one or the other, with the location being the only constant. Thus, the key to ageing in place would be to maintain the balance between the person’s capacities (progressively diminishing with age) and the environmental pressures [12,13,14], an idea equally present in the extensive reflection on active ageing [15]. Consequently, the act of ageing in place implies the ability to remain in housing as one ages which is presented as a goal in itself [10].

Personal variables interfere with the trend towards ageing in place since the elderly are not a homogeneous group and individual diversity grows with age: age, area of residence, education, years living in the same residence, housing tenure (especially important in countries such as Spain), and of course, health. Taking these factors together, Phillips, Ajrouch and Hillcoat-Nallétamby [8] qualify the previous definitions: ageing in place refers to the process of growing old without having to move from home, with the aim of maintaining continuity in the life course in a situation of independence. 

Ageing in place is therefore a decision, not the absence of a response, and consequently should be considered a strategy. Ageing in one’s own home is a way of appropriating one’s own old age and constitutes the most positive way of facing a life stage characterised by negative beliefs and the absence of prior socialisation at what constitutes this stage. However, it can be, in many cases, a decision with unwanted or negative effects, because if the demands of the environment exceed the capabilities of the person or vice versa, the person–environment relationship can lead to a situation of maladaptation [10,16], an issue which has been the subject of extensive analysis by environmental gerontology [12]. If the space does not allow an adequate adaptation of the older person, it will limit their social interactions, will produce negative consequences on their health [17], and will become a source of stress [18]. When the negotiation between the characteristics of the space and the person’s own needs and physical capacities is negative for the person, a lesser or ambivalent place attachment will develop, contributing to greater isolation and producing a very negative effect on the self-concept. Eventually, it may produce an undesired mobility (expulsion) from the urban space in which they wish to live [16,19]. Furthermore, in spite of the positive aspects of staying at home during old age, it is also true that there are elderly people for whom staying in their environment is a source of insecurity, isolation, and dissatisfaction [2,4,20] and can lead to various health problems, since not all the starting conditions are good [20]. As pointed out by Gilleard, Hyde and Higgs [21] ageing in place is an ambiguous position, signifying rootedness as well as rigidity, but the benefits of becoming “bound” to one’s own area, one’s neighbourhood, may be balanced by the costs arising from such “immobility”.

Despite these undeniable problems, generally speaking, ageing in place is probably the largest decision on ageing and its significance. Even when housing conditions are not adequate, the elderly choose to deal with the cost of the mismatch between their housing and their needs rather than move [16,19], since the home represents for most of them the private space of family relationships, with which memory and life memories are associated [22,23].

This statement of intent is a revolution in what old age means. The idea of frailty and dependence is rejected: most elderly people want to be autonomous. Even in the case of difficulties in basic activities of daily living (ADLs), they wish to remain in their own homes, and not join their children’s homes. In this regard, Rojo-Pérez [23] considers the place in which to grow old to be one of the most valued parameters of self-determination, especially when it is decided to reside in one’s own home to avoid burdening children or other family members [24].

### 1.2. Place Attachment

The link to the space and home in which the elderly have spent part of their lives, which translates into the preference for ageing in place, is referred to in academic literature as ‘place attachment’. According to Tuan [25], the place is a center of meaning constructed by experience. The consideration of the socio-physical space allows us to have a multidimensional and multidirectional conception of ageing [26,27,28] but also of the understanding of the process of inhabiting as a social construction that includes not only the physical, architectural, or administrative space, but also the experience of it. It is in these coordinates that the attachment to place is inscribed.

This attachment would include the emotional bond that is created between people and space. Rubinstein and Parmelee expressed it as the “set of feelings about a geographic location that emotionally binds a person to that place as a function of its role as a setting for experience’’ ([29], p.192). These feelings explain why people wish to maintain a relationship with the place to which they are emotionally attached and for which they feel some kind of affection, which in turn is reinforced by the time of residence in the place and the relationships established there [30]. Cross [31] summarises the extensive reflection on place attachment explaining its emphasis on the positive emotional relationship of people with specific places where they feel comfortable and to which they attach meaning. Nonetheless, the concept is of a dynamic nature, whereby changes in the life course or changes in place can disrupt the previous attachment.

Place attachment has a strong connection to place identity, which involves the incorporation of place into the broad concept of the self [32,33]. Place is fundamental in the processes of identity and self-definition of the self [33,34], becoming part of the social representation [35]. People’s identity is formed partly by their history and affinities with the places where they have lived [36]. It acquires a great influence in old age, being a key element in the quality of life and well-being [37,38] that contributes to shaping the identity in this vital stage [39,40,41]. This latter aspect is particularly important since the absence of socialisation and familiarity with the universe of old age [42] makes this period difficult to define and sometimes to cope with, from the self. Place attachment is related to the familiarity of space, whose meaning is not static, but transient and evolving, being able to change as people age [1,43]. In this research, place fulfills a relevant connecting function between old age and previous stages.

Rowles delves deeper into how older people develop attachment to space, analysing the changing relationship of attachment to place during old age, first in rural areas, and later in urban areas, with special attention to nursing homes, and making an interesting reference to the ontology of imagined places [43]. Rowles [37] investigates the subjective geographic experience of people over 65, distinguishing between two types of imaginary geographic experience, neutralising spatio-temporal distance: the reflective fantasy, which takes the elderly back to places in their biographical past, and the projective fantasy, which reflects the psychological link with current environments inhabited by close relatives, or with non-existent future worlds. This idea of geographical fantasy allows the appropriation of domestic space through the personalisation transferred by personal objects (the process in which the spaces became yours, turning the dwelling into a home). From Rowles’ studies and from the theory of the interaction between the person and his environment implemented by Lawton, the idea of place identity arises, which Proshansky conceptualised in 1978 as a specific component of self-identity (subidentity) that encompasses the knowledge of the places where the daily experience takes place, and which are as important as the social ones:

“Place-identity is defined as those dimensions of self that define the individual’s personal identity in relation to the physical environment by means of a complex pattern of conscious and unconscious ideas, feelings, values, goals, preferences, skills, and behavioural tendencies relevant to a specific environment” ([14], p. 147). Or, as defined later, a ‘‘potpourri of memories, conceptions, interpretations, ideas, and related feelings about specific physical settings, as well as types of settings’’ ([33], p. 60).

At the base of this structure is the individual’s “environmental past”, as well as the socially elaborated meanings referring to these spaces that the person has been integrating with his/her spatial relations. This “cognitive deposit” that configures the identity of place—of which, according to Proshansky et al. [33], the individual is not aware except when she feels her identity threatened—allows the person to recognise properties of new environments that relate to her “environmental past”, favours a sense of familiarity and the perception of stability in the environment, gives clues on how to act, determines the degree of appropriation or the capacity to modify the environment and, finally, favours a feeling of control and environmental security [34]. This spatial identity may entail more time than spatial attachment with sociodemographic factors being more important than time of residence [43].

The fact that most older people tend to have spent a long time in the setting in which they reside and their connection to the environment even in the face of loss of social participation [43] would explain the possible reluctance to implement change-and-break strategies (change of residence), even when the dwelling does not adequately meet their needs [16,19].

The cognitive deposit mentioned by Proshansky et al. [44] and its function as a defense when identity feels threatened allows a better understanding of the reasoning of ageing in place and how it connects with the appropriation of old age, in the sense that the person is immersed and involuntarily classified in a new category that establishes that at 65 years of age one enters a new protectable stage, associated with a series of negative valuations and losses of function. From these premises, ageing in place can function as an element of connection with the past life, with the known life, and, therefore, with one’s own self.

This attachment is potentially increased by the time spent in the dwelling and linked to the influence of emotional attachment derived from previous family experiences there [45]. Our house, our furniture, the way it is arranged, and the whole order of the rooms we live in, reminds us of our family and friends we are used to seeing in this environment [36]. As is well known, family ties in Spain are an essential determinant in the decision to remain in the residential environment. Even so, people who form single-person households also show a preference for remaining in their home, despite lacking that family background attached to the residential space, which should make us think about the reflection of self that is experienced in the living space at the individual level (as something that is his/hers, and that does not need to be shared for its experience).

In summary, the identity of place and attachment to the space would represent an added value to the dwelling in which the adult years have been spent before the entry into old age, which could explain the refusal to move from it even when that dwelling does not meet one’s needs or leads to adaptation problems and excessive pressure (in the sense indicated beforehand, and as demonstrated in [16,19,46]). 

### 1.3. Place Attachment in the Urban Environment

In this context, reflection on place attachment transcends the dwelling and encompasses the sphere in which people’s daily lives take place, understood as an area that extends beyond the residence itself [46]. 

European research on cities tends to show place attachment as a double dimension: social belonging (or the structuring of social relationships between residents in neighbourhoods, in different forms and degrees) and the link to place and as pointed out previously, the attitudes and ties which people feel towards the local environment they live in (for example, [47,48]). In addition to its relevance as a backdrop to social life, local space can be considered an active participant and an integral part of people’s social practices. According to Bennett [49], places are not just constructions or social imaginaries but also material things, articulated with the senses and with physical bodies. This is the reason why the author repeatedly uses the term “people scape” to allude to the many and varied connections within local areas through a shared world that includes acquaintances who are part of the place, as well as people whom one sees or speaks to only because they share a common space. In the materiality of places, the past, the present, and the future all meet—in buildings, in worn cobblestones, and in public playgrounds [49]. In this sense, the role that the local level plays in people’ lives deserve the greatest attention. 

In the sphere of urban local space, place attachment is strongly linked to relationships and social ties that the residents create in their neighbourhoods. Beyond family ties, these relationships can be of varying scope and depth. From strong bonds, linked to a more intense version of social capital [50], which encourage reciprocity and mobilise internal solidarity within groups, to more heterogeneous forms between individuals who are recognised as different, which fit under the idea of bridging social capital, relationships which, according to Rostila [51], can facilitate valuable links with people external to the group, activate distribution of information and encourage cooperative behaviours. This differentiation between bonding and bridging social capital has similarities with the distinction made by Granovetter [52] between strong and weak ties, which highlights the importance of the latter because of their capacity to establish connections that join people with social groups different from their own and provide them with information which is not normally available in their circle of belonging. In any case, loose and broad networks of acquaintance are very relevant [53].

These considerations constitute an aspect of the utmost interest in the reflection of Blokland [54] who investigates how local membership can adopt different versions in the contemporary city. In this line, she shows the importance of what she terms ‘public familiarity’, the social space constructed within a physical space through the participation and observation of daily interactions. According to this author, public familiarity is based on relationships, which, despite being temporary, influence the sociability of neighbourhoods and can help us develop a sense of inclusion or exclusion in a community through fluid encounters that repeat themselves [54]. The concept of the parochial realm that Lofland [55] takes from Hunter (1985), conveys a similar sense, in which neighbours and acquaintances feel involved in interpersonal networks, participating in a certain sense of community that provides physical and emotional security [24].

Within the framework of these coordinates, assuming the perspective of the elderly implies not only placing their personal and social experiences in the neighborhood, but also their own history [5]. There is no precise delimitation of what the elderly consider their place and their environment, although in the urban environment it often coincides with a subjective and experiential reinterpretation of what their neighbourhood is, not necessarily coinciding with administrative delimitations [19]. It is therefore a socially constructed subjective space charged with emotional meaning [43] reinforced by feelings of familiarity and the perception of safety and security [1,56,57]. Family plays a central role in this context since family location affects residential choices [56] and interaction with family members has a major influence on migration to the same local area [57].

Even if, for the elderly, the neighbourhood may gradually shrink to a “physical comfort area” in the housing surroundings [12], the space of the neighbourhood plays, in the case of women and old people, an essential role in the possibility of spontaneous interactions, and a basis for the development of social ties which can give people practical help and emotional support in specific moments [58,59]. In fact, Kearns et al. [53] highlight the results of one study of older women [60], in which a “sense of neighbourhood” score, that combined aspects of belonging and trust, was strongly associated with social support, while in another study, feeling less of a part of one’s community than before was found to be significantly associated with a transition into loneliness. The same was found by Prieto-Flores et al. [61] who showed that residential satisfaction had both direct and indirect effects on loneliness among older people, with relationships being stronger among those living in the community than among those living in care homes. In fact, remaining part of the familiar neighbourhood and community has physical and cognitive benefits [62,63] and impacts on self-esteem [39]. These factors and the importance attached by the elderly would explain their reluctance to leave their familiar spaces, even when certain large cities (e.g., Madrid) make “integrated ageing” very difficult (see [5,64] for a discussion).

### 1.4. Political Implications. Obstacles and Challenges in Achieving Age-Friendly Neighbourhoods

The physical and spatial context that influences people throughout the entire life course [65,66] becomes even more important during old age. They come to determine how we age and how we respond to disease, loss of function and other forms of loss and adversity that we may experience in this final stage of life [67]. Big cities are often characterised by problems such as the absence of green spaces and pollution, which will have an even more negative effect during old age, when the influence of the environment on well-being is greater. In fact, the elderly are more prone to suffer from urban contamination (NO_2_), lack of public space and noise pollution that increases vulnerability due to slower mental processing and sensory changes that take place in the ageing process (see a deeper analysis in [68,69]). Figure 1 shows a summary of the aspects that hinder the healthy development of the elderly in urban areas.

Beyond these challenges, for the influence of the environment to be positive in old age, it is necessary the urban space to be able to respond to the changing needs of older people and not become an impediment to their social participation, as this is a key component of health promotion [70]. When this does not occur, it not only diminishes intergenerational contact and its benefits, but it also limits the creativity of society [71] and threatens the integration of the elderly, thus increasing their risk of isolation as we have seen above. Loneliness and social isolation are associated with an increased risk of developing coronary heart disease and stroke [72], being also a risk factor for depression [73] especially in later life, and putting individuals at greater risk of cognitive decline and dementia [74] among other health and social problems [73].

As a consequence of these considerations, developing age-friendly environments has become one of the strategic objectives of the WHO’s Global Strategy and Action Plan on Ageing and Health with the purpose of adjusting the environment to its residents under the person–environment fit principle, a question already pointed at the Madrid International Plan of Action on Ageing (MIPAA), which addressed the challenge of constructing a society for all ages, focusing on three priority directions, one of which sought to secure supportive environments for older people to promote independence and empower older persons with disabilities to participate fully in all aspects of society. The Global Network of Age-Friendly Cities and Communities works in the same line of developing more age-friendly cities worldwide. This new discourse on ageing is therefore redirecting the debate on public policy regarding issues of social inclusion, commitment, and community development in conformity with the Age-Friendly Cities and Communities promotion [12]. In the US context, Feldman and Oberlink [24] developed a model of an “elder-friendly community” and a set of indicators to measure and help improve community capacity to promote the health and well-being of older residents. In the same way, in this context, Carvalho et al. [75] take from Hanson [76] the importance of creating and maintaining outdoor spaces that facilitate the elderly’s everyday life with public spaces of universal accessibility where they can develop their day-to-day activities comfortably, effectively, and safely [69]. The relevance of physical and social environments is therefore becoming a priority, while their assessment over the course of the years in different geographical spaces could offer key insights as to the actions of states in this regard. In this line, the recent advances in social services developed within the framework of Act 39/2006 (although not exclusively), which was designed to improve the quality of life of people with difficulties in carrying out basic activities of daily living and promote their autonomous existence at home, are also very relevant (see [77]). Something similar can be seen in other public and private strategies which, despite their shortcomings, lead us to speak of “the crossroads of care” [78]. Local and third sector initiatives in the framework of what we could call “the city of care” would be pointing to new possibilities to face the challenges of ageing in the city. However, these initiatives don’t have sufficient scope yet to enable the evaluation of their results in the medium term.

In summary, the academic evidence on place attachment shows the importance of older people’s desire to reside in their neighbourhood, a meaningful socio-physical space with which they are familiar and in which they can maintain their social networks and identities, an aspect that has begun to be taken up by policymakers and which will require the greatest attention in the future.

## 2. Materials and Methods

This article analyses how place attachment is shaped in an urban setting, using Madrid as an illustrative cultural case [79,80]. Madrid is the capital and biggest city of Spain (3,286,662 inhabitants in 2022), and therefore the one with the largest number of elderly people: 666,599 people are over 65 years old. The city has 21 districts which are further subdivided into 131 neighbourhoods, and are very different in terms of wealth and quality of life, but also from an urban perspective and in terms of social infrastructure, availability of green areas, urban fitment for old age or even pollution [5,68]. 

To obtain in-depth information, data were collected through ethnographic interviews with 37 people aged 65–95 and one focus group among the elderly aged 65–77 (total of 42 participants). The interviews and the focus group are understood as subunits that make up the case study. The people interviewed, relevant witnesses on facts of their personal experience ([81], were both men and women between 65 (minimum age required) and 95 years old. The interviews were conducted in the places of the interviewee’s choice, always in their neighbourhoods. Anonymised information on all participants can be found in the Appendix A.

Regarding the method of contacting the interviewees, cold-door contacts proved to be unsuccessful. The quality and depth of the information extracted this way are inferior to when it is done in trusted environments and with a previous work of approximation. Accordingly, contact with interviewees was made predominantly through children, grandchildren, and other relatives (mainly nephews and nieces), neighbours, volunteers, and neighbourhood associations. In some cases, the “snowball effect” was generated and the interviewees themselves provided new contacts.

The qualitative analysis, for which ATLAS.ti tool was used, began with the transcription of the interviews. During this process, which allows for contextualisation, the transcriptions were contrasted with the fieldwork notes collected during a previous process of non-participant observation. This avoids the risk that Valles points out of decontextualising the interview [81] so that during the development of the interview, the interviewer can follow the explanations of the interviewees. The “pencil and paper” paradigm was followed for the analysis when assigned with ATLAS.ti categories relevant to the research, following an open coding system [82] through the systematic active search for properties and the writing of memos (one of the ATLAS.ti’s most efficient tools-). This system facilitates analysis and representation, allowing, in turn, readjustments, demarcations, and deeper analyses with the potential to generate subcategories of analysis, following the axial coding defined by Strauss as “intense analysis done around one category at a time, in terms of the elements of the paradigm” ([82], p. 350) to allow a later integration of the categories and their properties. Once established, codes are assigned and the content is analysed according to such codes, grouping text fragments linked to each code and analysing separately each thematic module for the final elaboration of partial reports.

Regarding the ethical aspects of the research, the interviewees were assured that the information they shared would be anonymous, that their recordings would not be made available to others, and that the data would be published in such a way that they could not be identified. For this reason, transcripts of the interviewees, through which they could be identified, are not attached. Additionally, specific names referring to the specific location have been deleted. The results of the investigation were returned, and, in some cases, there was interest in receiving further information. This article will be sent to those people.

## 3. Results

As stated by previous research, place attachment is a strong predictor for older adults’ social well-being and social cognitive function [83] and any reduction in place attachment’s components could have adverse effects on their social well-being, a reason why place attachment should be considered as part of policies related to older adults’ neighbourhoods [84], elderly well-being and ageing in place. However, little is known of how it works. This section analysed this through the testimony of 42 people over 65.

### 3.1. What Ageing in Place Means—The Life Experience and Attachment

Ageing in place is a process that links residence, life cycle and old age, and although it depends on many factors (housing quality, personal finances, social support), it is based on a key aspect: attachment to space. This conception of space comprises two different but intertwined dimensions: that of the home (private space, associated with intimacy) and that of the neighbourhood in which it is located (social space), which is understood and experienced in many senses as an extended sphere of the home.

Even though home and neighbourhood are two different elements and the process of constructing both attachments could be influenced by different variables, with different weights and influences, it has been found that the reasons that explain the development of these two forms of attachment among the elderly (and therefore the desire for permanence) are closely related to each other. For some people, both attachments are constructed on the basis of the same reasoning, so that neighbourhood and home are understood in their imaginary as a single place fused in relation to past experiences, the time spent in the home and in the neighbourhood (considered as invested time).

*“I love my neighbourhood, and I’ve been here for 50 years, so I’m staying here”*.EM15 (68 years old).

*“If I won the lottery, I would never leave my house, I don’t know why… Because I like it, I like it. I like it (…) It’s probably because I’ve lived there for a long time. But no, no, I never wanted to leave”*.EM11 (68 years old).

In this regard, an illustrative example is found in the importance for the elderly person of the state and appearance of his or her home, as if it were a personal identification or “cover letter”. Thus, a person who has a clean house is a clean person. The cleanliness and condition of one’s home is, without exception, referred to with pride, as it is assumed to be a defining aspect of oneself. Having a house in poor condition is tantamount to being “defeated” by old age. Furthermore, not being able to have the house at the desired level of cleanliness (a totally subjective assessment) puts them under a lot of stress and, as we pointed out before, can be interpreted as the real entrance into old age:

*“Of course, I consider myself old when I have to climb to clean the windows… I can’t do it anymore. That’s because I also get dizzy. So I need someone to do the windows for me, to do a general cleaning”*.EM12 (73 years old).

The interviews and their analysis suggest that ageing at home implies an active decision, a strategy, even when it seems not to be the best option for the older person [16] or when there are better alternatives.

*“I like the neighbourhood (…) I wouldn’t, I wouldn’t go anywhere else. (…) We haven’t been able to put a lift in. I live on the second floor, so I have to go up and down. The day I can’t, I’ll put up with it”*.EM30 (87 years old).

*“Look, I have a flat in Villalba. It is very nice. Nicer than this one because this is a fourth floor without a lift. You can get an idea, 90 steps. Well, I don’t know. I go out, I think I have… the stones know me, I go out and… It’s not that I’m out of my mind, no. It’s that I like the neighbourhood more than anything else”*.EM31 (86 years old).

Ageing in place is the factual expression of a desire, which gives the person experiencing it a sense of continuity, as well as providing and reaffirming a sense of control over one’s own life:

*“I like to be independent. As long as I can (…) I would like to continue living, taking care of myself”*.EM4 (65 years old).

*“I tell you one thing, I wouldn’t sell my flat for anything because right now (…) never, I wouldn’t sell my flat for anything in the world, for nothing. I wouldn’t, this is my final home, until I die”*.EM9 (65 years old).

This chosen permanence in the familiar environment clearly contrasts with external definitions of old age characterised by loss and change of roles, a process that seems to be harder for men than for women.

*“As soon as you retire, you yourself think that you are no longer good for anything”*.EM1 (82 years old).

*“In the swimming pool there is only one man of our average age. In the previous group, none. (…) We, women, go much more to all the things… They play cards with each other. Yes. They go to petanque, yes. But then they don’t go to these centres to… They are ashamed or I don’t know why they don’t go”*.EM32 (70 years old).

*“I think that the adaptation of men when they stop working has nothing to do with us”*.EM35 (71 years old).


*“I retired two years ago, but I had been working for many years before that, with many activities, preparing for my retirement. And yet men finish working and it’s a step where they don’t know what to do. How do I fill my time, what do I do with my life? If they’re not attached to the missus… you must rely on them a lot. In general, right?”*
EM33 (67 years old).

Faced with loss of roles, remaining in the known space (where we are recognised) allows the elderly to experience old age as a stage in addition (and not juxtaposed) to the others. From this comprehensive framework, continuity in the known space makes it possible to connect old age with what (and who) one was before, making it possible to understand old age as a stage connected with the previous vital experience, so that the life cycle becomes a continuum. Old age can be thus delayed in one’s own imagination, as something associated with the loss of independence, which can occur to different degrees and for different reasons:

*“Oh, in my opinion, I’ve never been old. Because they always say “Look, poor woman, how old she is” and you say “But you have looked at yourself, what you are? “Oh, you’re… I don’t know how old you are, oh poor thing and so on”. “Oh, how she walks” and she says: “But you don’t consider yourself old” “Oh no, I don’t consider myself old”. [That was in the past]. And now I say, “Oh yes, I’m already very old, I’m already very old” (she laughs). You can no longer do the things you do, of course. But I’ve run the house. Everything, everything, everything, all the chores, and I’ve been doing everything, everything, everything. All the meals… well, an assistant comes once a week and she cleans everything for me, but the rest of everything, the ironing… Everything (…) I’ve taken care of. Well, that’s probably why I didn’t consider myself… I was still valid”*.EM5 (85 years old).

*“I see myself today as an old person who cannot do with my life what I want to do, because I don’t have the means to do it. So… (…). You can also be an old person like my mother was [she suffered a stroke], you can be left in that way that you depend on a lot of other people. Right now, let’s see, [Isabel] Preysler [socialité] is 71 years old and nobody tells Preysler that she is old”*.EM4 (65 years old).

Only when it is accepted that the experience of inhabiting is different and will never be the same again, will strategies of rupture be considered [16,19].

### 3.2. The Subjective Construction of the Neighbourhood

The neighbourhood is, altogether with the family, school, and work, one of the main spaces of socialisation, where individuals conform a large part of their social networks, limiting or favouring the type of potential social relations. The subjective idea of neighbourhood refers to an area considered homogeneous and familiar in the imaginary of the person who mentions it and may not coincide with the vision of another person. It implies reference to attributes based on physical geography, resident population, social capital, and commercial fabric, but above all it is delimited and reinterpreted based on the use that the person needs and makes of his or her surroundings. This is a dynamic perspective, as the neighbourhood is constantly evolving, due to the action of incoming and outgoing flows of people and resources, and even due to the urban planning itself, which can change over time.

In this subjective configuration of the neighbourhood there is an emotional involvement and an emotional investment, at the same time as both the home and the neighbourhood become a form of presentation of the person. In this sense, the history of the neighbourhood is the history of oneself. The space in which one lives symbolises a series of values, even personal qualities, reasons for which the residential and social value of the neighbourhood is exalted—be it for its beauty, comfort, historical value, economic value or for the future and possible revaluation of the neighbourhood, or even for its contrast with neighbouring districts:

*“Yes, this is the best neighbourhood here, the best place, the best area. We are left to our own devices, but… Yes, this is the best neighbourhood there is”*.EM16 (68 years old).

*“My sisters, one lives in [name of neighbourhood] and the other in [name of neighbourhood], in some new houses they built there. What a difference! We used to come here, and this was “life”. Here you always see people in the street. (…) And I like meeting people. And not [being] in a lonely neighbourhood, which is where you can get scared”*.EM34 (77 years old).

Among the people interviewed, there is also a hypothetical resistance to leaving the neighbourhood based on these reasons. Part of this resistance, and of the pride with which the neighbourhood is spoken of, can be found in the identity dimension, which is manifested through the aforementioned discursive defence of the neighbourhood, which is presented more from subjective parameters (what it means to the person to live in that neighbourhood) than from objective ones (what the neighbourhood is really like), so that the positive aspects are exalted and the negative ones are minimised—or ignored.

### 3.3. How Is Place Attachment Constructed? Influencing Factors

“When I speak of home, I speak of the place where those I love are gathered together”. Dickens, 1839.

#### 3.3.1. History and Family

The identification of staying in the desired place with autonomy and as a form of resistance to a definition of old age—in which one has not participated, external to the ageing person—although with different manifestations, has been shown in one way or another in all the interviewees, regardless of gender. Even when the housing and/or neighbourhood conditions are not adequate, and even when objectively there are problems, the positive aspects are always highlighted [85]. Within the discourses, these positive aspects tend to be related to their own historical experience and contributions to the community (e.g., being part of the neighbourhood association that managed to get an emergency shelter, having promoted the installation of a lift in the building or necessary changes for the improvement of neighbourhood life).

In addition to the time spent in that space, different factors that intervene in the construction of attachment among the elderly and reinforce the desire to continue have been detected, such as the family and the household memories, understood as a form of connection with a past that continues to have meaning in the present moment. 

The influence and importance of the family institution in place attachment can be rooted in the potential social support that family members give each other. Research has shown the powerful force of family ties as a determinant of the decision to stay locally. Hedman [56] finds a clear inclination to choose residences where other family members live. In her research, she discusses the affinity hypothesis, according to which, people may move near family members simply because they like them and want to live near them. A second hypothesis, the facilitating hypothesis, is complementary to the previous one, where it suggests that family members may offer potential help of several kinds depending on how closely they live. For instance, family can play, among others, facilitating roles in exchange of childcare or daily household duties. Grandparents, for example, are important providers of childcare, particularly in Spain [86]. Both affinity and facilitating hypotheses emphasise that family affects location choice directly, that is, the destination is chosen because of a wish or need to live near family.

*“My brother came, then my mother and then we came. My children also live nearby”*.EM6 (66 years old).

Proximity to the extended family (especially to relatives in the ascending line) is an important factor in the choice of residence, as the elderly stated in their interviews. As underlined by Mulder and Malmberg [57] individuals whose parents live nearby are less likely to migrate to other local areas than others. In addition to attachment, it allows for a more continuous manifestation of family solidarity leading to closer intergenerational and even intragenerational relationships. Tobío’s research [87] revealed that grandmothers’ support was clearly associated with the spatial proximity of their daughters and grandchildren; three out of four working mothers whose mother lived in the same house, building or street were helped by her in childcare ([87]. It is therefore common to find members of the same family in the same neighbourhood and even in the same building (extended co-residence) [19]. 

*“We bought a flat near my mother-in-law (…) she lives a few streets away, on her own (…) she is 92 years old. Before, she used to come by herself to eat, but now she has to be picked up by car (…) and she comes every day to eat”*.EM19 (67 years old).

*“My husband was born in this house. And it [the neighbourhood] attracted him a lot. And my mother-in-law said: “Well, come here”. My children have grown up here, in this neighbourhood (…) I like everything about the neighbourhood (…) I like, that’s why I tell you: I like everything about the neighbourhood”*.EM34 (77 years old).

Concerning household memories, they have more to do with a connection to one’s own family history. The family home constitutes not only a physical space but also a social, emotional and attachment space, in line with the environmental past suggested by Proshansky et al. [33]. Sometimes it is also a link to one’s own history and that of one’s ancestors, as a form of continuity (legacy) in space that is linked to the perspective of the life course and belonging during old age [19]. 

*“My youngest son was born here. The older ones weren’t, but they came when they were very little. And so, there’s still that warmth, isn’t there? And… I don’t know. There is still the warmth of the family, and of the children and so on”*.EM8 (65 years old).

#### 3.3.2. Recognition and Identification

The family factor is not the most important nor the only intervening aspect in the construction of place attachment. In fact, the way in which the person feels recognised as part of the neighbourhood (endogroup, sometimes made up of intimate strangers interrelated with each other by weak ties) will be key. Mutual recognition as part of this group, in the subjective space of the neighbourhood (where I am known and where I know, of which I feel part), fulfills a fundamental function, which is to maintain the connection with identity and reinforce the self-concept, but it is also experienced as a continuity with the life course: it is me, only older.

If retirement or the children´s leaving led to a potential loss or change of roles, remaining in the neighbourhood and being recognised (even superficially) as part of a community helps to ensure that old age does not represent a rupture or a very different stage. Continuity in the known environment makes it possible for old age to go from being an identifier (because outside your environment, where you are not known, you are “just” an old man/woman) to being a dimension added to others that make up the identity, but not necessarily the most important one: Juan outside his neighbourhood is an old man (one more) but in his neighbourhood, Juan is Juan, or maybe “the old Juan”, but he is not just an ordinary old man.

*“I’m so happy in my house and everything, in my neighbourhood. Do you know what it’s like to go out on the street and bye-bye! Hello! Everyone knows me here: How are you, how are you? I was born here!”*.EM24 (67 years old).

Continuity is especially relevant in the recognition that decisions in old age are shaped by previous experiences, highlighting a continuous dimension with the past and recognising the conditioning factors that result from lived history: “Later life transitions are influenced and shaped by earlier experiences, and they, in turn, shape the subsequent life course” ([88], p. 591).

This dimension of continuity and recognition contributes to greater participation (as part of the group, no matter how unstable it may seem) and fosters social relationships. Continuity in the social participation of older people in accordance with their needs, desires and capabilities is important for active ageing.

*“I am very happy. Because there are many activities, you meet many people, you talk to many classmates… So, I am very happy. Yes, yes, yes, yes (…) So, very good. And I tell you, I am very happy here [local community center], that’s the truth”*.EM36 (68 years old).

Particularly relevant is the influence of the temporal aspect, which would affect all the other elements on which attachment is built, and which has a decisive influence on the appearance of the dimension of identity, continuity, and recognition, which fundamentally refers to the feeling of knowing oneself to be recognised. This facet impacts one’s sense of identity and allows the connection of old age with the rest of the life cycle.

*“Yes, because here all the neighbours have always known each other”*.EM10 (92 years old).

*“Yes, yes, we know each other. All the old people know each other. When one of us dies, I say: “Oh, what a pity”. Of course, if it’s… Of course, this neighbourhood is not a neighbourhood, this neighbourhood is a village”*.EM37 (66 years old).

#### 3.3.3. Ontological Security and Microgroup Solidarity

Another of the operating factors points to the dimension of perceived safety, which is conceived as the absence of danger and as psychosocial well-being, in terms of the urban setting being free of unfamiliar spaces and environmental stress, but also of social stress:

*“I feel at ease, I am in my shelter”*.EM25 (70 years old).

*“I want my neighbourhood because I know a lot of people here. We all know each other, we all know where we all stand, we all know where I stand, and we all know each other”*.EM15 (68 years old).

This does not necessarily imply close relationships or friendship, but rather the concept of intimate strangers would apply. According to this concept, even if neighbours do not know each other personally or have not exchanged conversation, they have developed a certain degree of public familiarity (in the sense indicated by [54]) because they are part of the same group. This level of relationship is consistent with the atmosphere of comfort (comfort zone) that is sometimes created in neighbourhoods [89], also linked to the potential for exchange that is derived from weak ties or absent ties (such as daily greetings) [52]. This feeling of familiarity is directly linked to the residential stability of the population, which is still present but at risk in certain neighbourhoods due to processes of touristification and gentrification [46].

Among intimate strangers it is easy to find forms of collective organisation and group solidarity in situations of need, but also greater social control. Precisely, it is this greater social control, without denying the negative consequences that sometimes occur, that comprises one of the elements that contribute to people feeling comfortable and safe. Somehow, the behaviour of the other (part of the same group, not explicit but recognised—neighbours, neighbourhood-) is predictable, and even when intra-group distinctions are created, belonging is exalted.

*“Many times, I go by myself. (…) I don’t have prejudices. I deal with my butcher, with my greengrocer, whose name is Julito (…) And I have other friends that we see each other, we meet (…). They are acquaintances but we see each other. I have lived with them, and we still deal with each other”*.EM 37 (66 years old).

This public familiarity could equally be defined as a form of social capital that implies that residents have enough information from everyday interactions to recognise and classify other people. For this reason, a potential change of housing or neighbourhood would have a great impact at the psychosocial level [19]. This subjective security decreases in situations of instability in the social fabric [46], and especially in environments with a large number of vacant premises or dwellings or where the dwellings are intended for uses other than residential [19].

*“No one is living next to my door. The three years I’ve been here, that house is uninhabited, and nobody live in it (..) If anything happens to you, you go out there, you scream, whatever, but I have no one and I am panicked”*.EM20 (75 years old).

*“I liked the way it was before more than now (…) Well, we were all, as they say, as if we were family. The people from the neighbourhoods stayed longer, more people lived there, more time”*.EM31 (86 years old).

*“The neighbourhood has changed a lot. In terms of neighbourhood, I can tell you, for example, in my house before we were all… well, families. And, very close-knit families, that’s the truth, which is not the case now. That has been lost because now most of the apartments have been sold and are rented out”*.EM36 (68 years old).

The two previous dimensions are in turn related to the capacity to develop micro-group solidarity:

*“They already know you in the neighbourhood. Because you can fall and say: “Look, [name] has fallen” … (…) “So-and-so, your mother is here” or “Mengano”, or whatever, or the neighbour who knows him. That is fundamental (…) Yes, because you need the support of the neighbourhood, and normally if you have lived there, you have friends in the neighbourhood because you know each other”*.GD1 (65 years old).

A clear example of this is the exchange of house keys among neighbours, a practice that implies the psychological security that someone will be able to get in, should there be an emergency [46]:

*“My neighbour Paquita has them [home keys], yes, and now I have hers”*.EM31 (86 years old).

The collective responses to the situation caused by the COVID health emergency and the neighbourhood organisation in the face of the isolation problems resulting from the lockdown effect have been the clearest example of the capacity to develop micro-group solidarity among intimate strangers in situations of need:

*“The guy living up here, the one who left, used to bring it [the groceries] up to me. He left it for me here at the door, and we didn’t see each other or anything (…) Before that, I didn’t even know him or anything”*.EM29 (77 years old).

In the discourses there are also different references to community life, although they imply different degrees of depth and involvement, recognising the value of social infrastructures (in the sense pointed out by Klinenberg [59]) such as libraries, social and senior centres, churches and businesses.

## 4. Discussion

This article, built on qualitative methodology, delves into the importance of place attachment in an urban space as a constitutive part of ageing in place. To do so, we have analysed the experiences and perspectives of 42 people over 65 years ageing in place in the city of Madrid. 

While it is important to keep in mind the divergence of applications and to understand that the concepts associated with place identity or ageing in place are nuanced and somewhat elastic, the novelty and interest of this study lies in the contribution to the understanding of ageing in place, as a mechanism that allows elders to somehow “own” the environment in which they age, connected at the same time with the previous view of attachment and place identity, and assuming the understanding of home, regardless of its characteristics (number of members, composition) and changes in space.

As we have seen, place attachment has two dimensions: attachment to one’s own home (private sphere, associated with intimacy) and to the neighbourhood (social space, understood as an overlapping and extended dimension of the home). In fact, there is a continuity between one’s own home and the neighbourhood because an emotional investment and a feeling of familiarity and belonging is projected onto both.

We have verified the importance of family ties in shaping place attachment, a particularly relevant aspect in Spain, insofar as they provide different—and often essential—support resources. The coincidence of family members residing in the same neighbourhood allows for a more continuous manifestation of family solidarity leading to closer intergenerational and intragenerational relationships. In a more immaterial sense, household memories, with their great emotional content, allow us to relive and project the moments lived in the past onto the experiences of the present moment, thus reinforcing the links with the place attachment.

Similarly, the qualitative approach allowed us to understand that staying in familiar surroundings provides security, understood as the absence of perceived danger, and mitigates the effect of the processes of rupture with respect to life experience (what is known, who we are and what defines us) that can occur in old age with the change of roles, especially among men, as some of the interviewees referred to. We have also verified the importance of the weak (and absent) ties that foster the generation of a certain atmosphere of comfort among the intimate strangers who live in the neighbourhoods, by making it possible for the environment to be free of unfamiliar spaces and environmental stress, all of which not only reinforce the feeling of security and well-being but can become active and produce forms of collective organisation and group solidarity. The great relevance of this public familiarity, however, can break down in situations of instability and turnover in the social fabric of neighbourhoods.

Considering all the aspects reviewed, even if family roots (past or present), community connections and expressed satisfaction with the neighbourhood are significant predictors of attachment to the space, the mechanism that contributes to the construction of the self appears as the truly determining element. The reinforcement of identity that comes from not being disconnected from the life cycle and the idea of continuity emerge as authentic pillars of place attachment. Remaining in the known environment where one is recognised makes it possible for the dimension of ageing to transmute from being an identifier of the individual (because outside the environment where you are not known, you are seen as “that old person” and the age become the preponderant identifier) to become one more characteristic of identity that is added but it is not necessarily the most important one. In this way, it fulfils a fundamental function, which is to provide continuity with the life cycle through external recognition.

The figure below (Figure 2) represents a synthesis of how this social and emotional meaning of attachment can be structured from the qualitative findings by independent but mutually influencing sub-dimensions:

Place attachment is built on housing and neighborhood (two spaces interconnected by the housing entrance and marked by uses and subjective experience) and is the result of dynamic processes influenced by a number of dimensions interconnected to a greater or lesser extent. Without expressing an order or strength, these would be the following: (i) the family, as we have expressed it and along the lines of the potential family solidarity that it can offer at the level of the whole life cycle (and in a bidirectional way); (ii) the symbolic dimension acquired by the experiences at home, generally connected to the family that we have created or the one in which we grew up; (iii) the subjective perception of security that one has about the home and the neighborhood (also understood and subjectively constructed); (iv) the solidarity and social support that the social fabric of the neighborhood can offer (manifested in many different ways and with different degrees of intensity and depth); (v) the social identity dimension, understood as the continuity of the role as a neighbor in the context of a life stage characterised by the potential loss of other roles, and the social recognition (by peers, in this case neighbours) of this social identity; (vi) one’s own history lived in the environment, which is a psychosocial construction based on personal experience in the neighborhood throughout life or, at least, in the years prior to old age.

All in all, according to the findings, ageing in place is highly functional for the ageing person. In this regard, as we have seen above, it is key to develop appropriate measures that guarantee the permanence of the elderly in the environment in which they wish to grow old, participating in the life of their communities, for as long as possible.

## 5. Conclusions

The main conclusion is that growing old in familiar surroundings mediated by place attachment plays a fundamental role, embedded in the identity and security offered by the home and familiar surroundings, but, above all, it symbolises the capacity for independence and autonomy.

In this line, the results reveal that staying in the familiar environment provides perceived security, which is deeply related to the construction of place attachment, group solidarity and belonging. Therefore: -When ageing in place is mediated by place attachment, the associated sense of belonging mitigates the effect of the processes of rupture with respect to life experiences that can occur in old age with the change of roles associated with retirement and other life processes.-Ageing in place thus makes it possible to experience old age as a part of the life cycle, a stage that can be understood as an ongoing reality to which years are being added. 

In view of the results, an inevitable conclusion is the necessary appeal to public policies to implement measures that have an impact on the construction of a society that ensures safe environments for the elderly in line with plans such as those derived from the age-friendly cities and communities’ networks. Linking this idea to the social space of neighbourhoods, it is also necessary to implement actions that not only create spaces for socialisation in which the elderly feel comfortable and safe, but also reduce the impact of resident turnover.

The analysis presents an approach to the process of attachment in old age. However, there are limitations to the research. The use of the case study allowed for the participants’ real perspective to be approached and thus to understand the object of study, pointing out the complex relationships between the phenomena that constitute its context. However, the case study allows for a very limited generalisation of the results obtained. Another limitation of the study is related to the fact that the results are conditioned by the specificities of Spanish society, particularly regarding family ties. Potential extrapolations to other realities should take this point into account.

In addition, despite the coincidence in the responses of the elderly interviewees that the article details, it is necessary to consider that the study does not take into account the differences in depth that exist between the Madrid neighbourhoods. Considering this last point, future work, with a larger sample size, will make it possible to include socioeconomic variables and compare the processes between different neighbourhoods. Assessing other cities in terms of comparison and considering new dimensions (such as, morbidity, socioeconomic aspects, use of spaces) would be of great interest as well as considering specific indicators of active ageing and quality of life. Another very interesting future line of research would be the comparison between urban and rural environments. Finally, another line of research, already underway, is aimed at understanding the role of alternative homes and neighbourhood support services (within the conceptual framework of “the city of care”) for improving the quality of ageing in place and its role in place attachment of the elderly.

## Figures and Tables

**Figure 1 ijerph-19-17052-f001:**
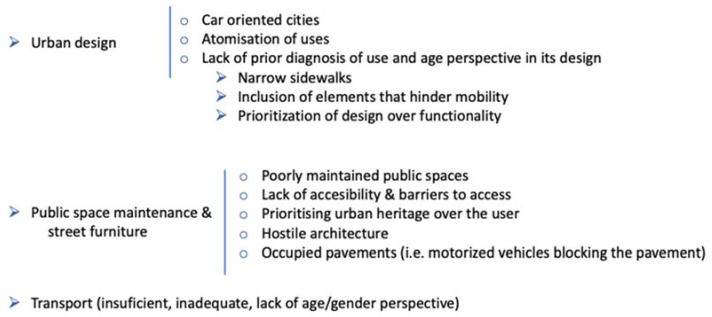
Problems identified in the process of integrated ageing in cities. Source: adapted from Lebrusán ([5], p. 372).

**Figure 2 ijerph-19-17052-f002:**
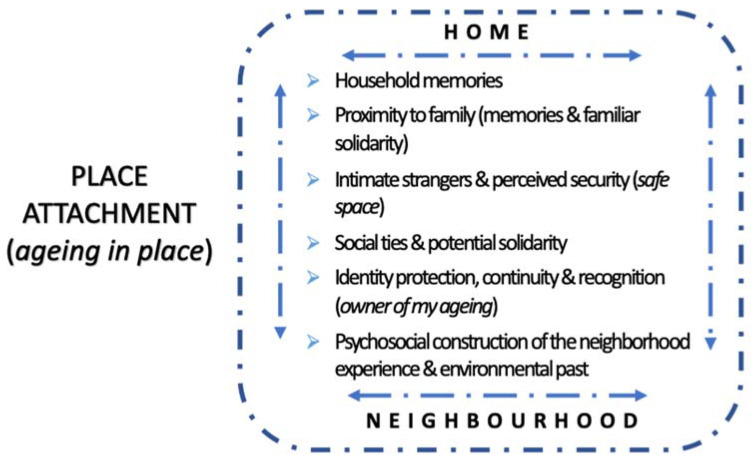
Factors influencing place attachment in ageing in place. Source: Prepared by the authors based on the qualitative analysis.

## Data Availability

Not applicable.

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
