# Peer review of "The Importance of Place Attachment in the Understanding of Ageing in Place: “The Stones Know Me”"

_ijerph, 2022, doi:10.3390/ijerph192417052_

Round 1

Reviewer 1 Report

Dear Authors,

The present work aims to assess the meaning of place attachment and its importance to the elderly population in urban areas. I enjoyed reading this case study.

The research topic reveals to be very actual and interesting. The literature review seems appropriate for analysing proposed constructs, such as ageing in place, place attachment, and place attachment in an urban environment.

The methods used to collect data and the number of interviews (42 participants) appear to be sufficient.

Below, we'd like to provide some comments and suggestions. These aspects should be addressed as it's essential to the structure of this paper.

1 - The abstract does not clearly state the results of this paper. Some improvements could be made, such as focusing on the findings and results of this study or its limitations.

2 – Within the literature review, to fully access these constructs, it would be advisable to pay some attention to the construct’s age-friendly environment or communities, as there is extensive literature on this matter. The present literature review focuses on the urban environment but fails to relate both constructs. Some literature about these constructs should be added.

For instance, Feldman & Oberlink (2003), in their work The Advantage Initiative: Developing community indicators to promote the health and well-being of older people, and Bild  & Havighurst, R. J. (1976) in their work Senior citizens in great cities: The case of Chicago, are some examples of authors that have conceptualised the construct  - age-friendly environment.

3 – Baring in mind that there is an essential gap in the literature review, once this problem is solved, these constructs must be incorporated into the discussion, as they might alter the results and implications.  Also, it would be interesting to contrast methods and results with similar studies mentioned in the literature review.

4 – Although the abstract mention “the processes whereby place attachment is constructed by the elderly and the role of the experience of neighbourhood are little known” in the discussion section, little attention is made to this dimension of the proposed model. I couldn’t find references to this dimension on the proposed factors influencing place attachment in ageing in the place model proposed on page 14.  

You have mentioned two different but interconnected dimensions: the home (private space) and the neighbourhood (social space). Also, you mentioned on page 7 that home and neighbourhood are two different elements.  

My question is: Which concepts should be included in the model on this dimension? If you take another look at the data, you might find the link to this. For instance, on page 5 and 6, you mention a previous study where the concept of “sense of neighbourhood” is explored. Furthermore, you have said, the existence of 131 neighbourhoods in Madrid and that the interviews were always conducted in their neighbourhoods.

5 – Along the text, 38 references of “neighborhood" are found, and it should be written neighbourhood. Please correct this typo.

I hope these comments can help you to improve your work.

With my best regards.

Author Response

Dear reviewer 1,

We highly appreciate the comments and suggestions with regard to our manuscript ‘The importance of place attachment in the understanding of aging in place: “the stones know me”’(ijerph-2061007). We have made in-depth corrections to the text as you suggest, paying due attention to each of your observations, as you will see below. All corrections have been introduced into the text with change control, so that changes can be detected more easily. Please let us know if you feel more changes should be made. We hope you find these changes satisfactory.

1. The abstract has been modified in the sense you required. We have added 2 paragraphs where we focus both on the main finding of the research and on one of its potential limitations.

2-3. In order to include the issues you mentioned, the article has been restructured and a new subsection [1.4. Political implications. Obstacles and challenges in achieving age-friendly neighbourhoods] has been added to summarize the additions. The new organization facilitates the understanding and integration of the previous contributions and additions in this new version.

Furthermore we have found both Felman & Oberlink’s and Bild & Havighurst’s texts very useful and included their quotations throughout the article.

4. The discussion section now includes explicit mentions to place attachment and experience of the neighbourhood. We have also modified the proposed model making it more comprehensive. A paragraph explaining the model and the different components is also added in order to illustrate some of the issues discussed in the graph.

5. The word ‘neighborhood’ has been changed by ‘neighbourhood’ throughout the whole article.

Reviewer 2 Report

Comments

The aging in place concept requires an interdisciplinary approach led by gerontology, valuing interventions at different scales: national, regional, community and individual.

In economically more favored countries, when the elderly begins to lose autonomy and capabilities, the option is often institutionalization, while in economically more fragile countries, aging in place appears not as an option, but as a necessity, given the limitations of the systems of care. social security and the lack of institutional alternatives.

Article is focused on Madrid and compares private space with social space. By the elderly and the role of the experience of neighborhood are little known, as the authors say. Aging at home aims to systematize and highlight models and strategies implemented in any country, which fit the principles of well-being and quality of life of elderly people, especially those who live more isolated.

Suggestions

The article should highlight more the importance of aging in place in functional and emotional terms, in order to keep older people not only living in their homes, but also participating in the life of their communities, for as long as possible. .

The authors should go further in terms of acknowledging the importance of home support services, drawing attention to initiatives that, due to their innovative nature, can effectively constitute good practices.

The study should add value and call for the reinforcement of installed capacities and the training of the teams responsible for these projects, thus contributing to improve the impact on the quality and scope of ongoing interventions.

Another suggestion is for the team of authors to find out to what extent the place where people live is responsible for the way they age. When assessing the extent of post-traumatic stress in the elderly, community cohesion represents an important source for overcoming the adversity situation.

However, another theme concerns the perception of the lack of security in the neighborhood, an issue that can aggravate depression. Social vulnerability restricts physical activity, fueling a vicious circle. Chronic diseases are also related to depression, as they limit the person's functional capacity and willingness to interact socially. If the environment is hostile to any type of exercise, even a simple walk, the health condition will only tend to get worse.

For future investigations, I point out that life expectancy has doubled over the last 150 years as infections, vaccines, nutrition and maternal health have radically improved, if obvious dangers like disease or stressors don't get in the way. Today's big killers – namely cancer and dementia – are much harder to tackle (Nature Communications (2001), “Longitudinal analysis of blood markers reveals progressive loss of resilience and predicts human lifespan limit)”. This is a problem that can inspire authors for future work, preferably multidisciplinary.

Author Response

Dear reviewer 2,

We highly appreciate the comments and suggestions with regard to our manuscript ‘The importance of place attachment in the understanding of aging in place: “the stones know me”’(ijerph-2061007). We have made in-depth corrections to the text as you suggest, paying due attention to each of your observations, as you will see below. All corrections have been introduced into the text with change control, so that changes can be detected more easily. Please let us know if you feel more changes should be made. We hope you find these changes satisfactory.

Reference has been made to the different suggestions throughout the revised text, introducing new bibliography and connecting the themes proposed by you with another research already carried out.

In addition, in order to include the issues you mentioned, the article has been restructured and a new subsection [1.4. Political implications. Obstacles and challenges in achieving age-friendly neighbourhoods] has been added. We have tried to underline the points you made in this new subsection. We think the new organization facilitates the understanding and integration of the previous contributions and additions in this new version.

We have included the future lines of research on which we are working at the moment.